# Phytotherapeutic, Homeopathic Interventions and Bee Products for Pediatric Infections: A Scoping Review

**DOI:** 10.3390/nu17193137

**Published:** 2025-09-30

**Authors:** Camilla Bertoni, Ilaria Alberti, Niccolò Parri, Carlo Virginio Agostoni, Silvia Bettocchi, Stefania Zampogna, Gregorio Paolo Milani

**Affiliations:** 1Department of Veterinary Sciences for Health, Animal Production and Food Safety, University of Milan, 20122 Milan, Italy; 2Pediatric Area, Fondazione IRCCS Ca’ Granda-Ospedale Maggiore Policlinico, 20122 Milan, Italy; 3Department of Emergency Medicine and Trauma Center, Meyer Children’s Hospital IRCCS, 50139 Florence, Italy; 4Department of Clinical Sciences and Community Health, University of Milan, 20122 Milan, Italy; 5Department Pediatrics, Azienda Sanitaria di Crotone President of SIMEUP (Italian Society of Pediatric Emergency Medicine Urgency), 88900 Crotone, Italy

**Keywords:** infections, children, homeopathy, phytotherapy, immunostimulants, natural remedies

## Abstract

**Background**: Acute infections in children are prevalent and often lead to antibiotic overuse due to the lack of evidence-based alternative approaches. Phytotherapeutic, homeopathic treatments and bee products are frequently sought as alternative or adjunctive therapies. This scoping review aims to map the existing evidence on the efficacy and safety of these interventions in managing acute pediatric infections. **Methods**: A comprehensive literature search was conducted across multiple databases to identify studies assessing the use of phytotherapeutic, homeopathic remedies and bee products in children with acute infections. Gastrointestinal infections were not considered since the use of non-antibiotic treatments (probiotics) in these conditions has been widely addressed. **Effectiveness**: Phytotherapeutic agents and bee products demonstrated promising results in reducing symptom severity and duration in respiratory infections, whereas homeopathic data were limited and inconsistent. Regarding safety, both interventions were generally well-tolerated, with few adverse events reported. No studies or very limited evidence were available for other acute infections such as urinary, dermatological, osteoarticular and nervous system infections. **Conclusions**: Phytotherapeutic interventions and bee products, particularly in acute upper respiratory tract and acute bronchitis, show encouraging signals of efficacy and safety in pediatric populations. However, evidence for their use in other frequent childhood infections, such as otitis media, or gastrointestinal infections, is almost entirely lacking. In addition, the available literature on homeopathic remedies is scarce and methodologically inconsistent, preventing any firm conclusions. Well-designed, large-scale clinical trials focusing on these underexplored conditions are needed to clarify the potential role of phytotherapeutics and homeopathy in pediatric infectious diseases.

## 1. Introduction

Acute infections are highly prevalent in childhood, encompassing a broad spectrum of conditions affecting various organ systems [1]. These infections pose a significant burden on both public health and healthcare systems worldwide and are typically caused by viral agents, although bacterial etiologies can also play a relevant role. Among these, dermatological infections [2], urinary tract infections (UTIs) [3], upper respiratory tract infections (URTIs) [4] are particularly common. Also, gastrointestinal (GI) and central nervous system (CNS) infections are particularly concerned due to their frequency and potential severity, respectively.

Acute pediatric infections present a diverse array of symptoms, including irritability, discomfort, and disruption to daily routines. These manifestations not only affect the child but also significantly impact parents and caregivers. Parents sometimes experience increased distress, worry, and anxiety when their child is ill and this emotional burden can lead to disruptions in family life, including changes in daily schedules and social activities [5,6]. While most of these illnesses are self-limiting, their management primarily focuses on symptom relief and preventing complications. Conventional pharmacological treatments, including antipyretics, analgesics, and antibiotics for bacterial infections, are commonly prescribed [7,8].

Nutraceuticals and dietary supplements might be potential adjunctive therapies in the management of these common pediatric conditions. According to the European definition, phytotherapy is the “science-based medicinal use of plants and preparations derived from them, in the treatment, alleviation, and/or prevention of disease or injury, according to recognized standards of quality, safety and efficacy” [9]. Natural remedies encompass various therapeutic approaches, including homeopathy, which derives from the Greek words “homoios” (similar) and “pathos” (suffering). This system of medicine is grounded in the principle of similars: the idea that a substance capable of inducing symptoms in a healthy individual may stimulate the body’s self-healing mechanisms in someone exhibiting similar symptoms [10]. These substances may possess immunomodulatory, anti-inflammatory, antimicrobial properties, offering potential benefits in alleviating symptoms, supporting immune functions, and potentially reducing the duration of illness [11,12]. Also, we have introduced a “third” category including bee-derived products, since these cannot be classified as phytotherapeutics (given their non-plant origin) nor as homeopathic remedies. These remedies cannot be classified as phytotherapeutic, since they do not have a plant origin, nor as homeopathic, as they are not obtained through dilution and dynamization processes. For this reason, honey, propolis, royal jelly, and other bee products are commonly referred to within the framework of apitherapy, which represents a distinct category. Despite their growing popularity and inclusion in integrative pediatric care, the efficacy and safety of phytotherapeutic, homeopathic remedies and bee products in children remain a topic of ongoing debate, as many products lack robust clinical validation and the quality of the available evidence is often inconsistent.

Recent concerns about antibiotic resistance, side effects of pharmacological treatments, and a growing preference for natural healthcare solutions have spurred interest in the use of homeopathic, phytotherapeutic alternative remedies and bee products for pediatric acute infections. A recent systematic review and meta-analysis [13] examined the role of minerals and vitamin supplementation in respiratory infections, revealing limited efficacy, with only modest benefits observed in certain contexts. However, the analysis primarily focused on respiratory infections and did not consider other common pediatric infections. It also excluded phytotherapeutic and homeopathic treatments.

This scoping review aims to synthesize the current literature on the use of phytotherapeutic, homeopathic remedies and bee products in managing acute infections in children. Specifically, it aims to evaluate the state of the art on the interventions in pediatric populations with acute infections. Furthermore, it aims to summarize the efficacy, safety, and mechanisms of action of these remedies in common pediatric illnesses, comparing them to conventional treatments or placebos. Finally, the review seeks to identify gaps in the literature about the efficacy and the rationale use of these nutraceutical substances in acute infections in children and propose directions for future research on the role of nutraceuticals in pediatric acute infections.

## 2. Materials and Methods

This study was initially conceived as a systematic review (PROSPERO, CRD42025639694). However, during the exploration phase and after a preliminary analysis of the available literature, it was deemed more appropriate to adopt a scoping review approach, as the evidence was too heterogeneous in terms of populations, interventions, study designs, and outcomes to allow for formal systematic synthesis. Additionally, data were entirely lacking in several relevant areas. The scoping review thus enabled us to map the breadth, nature, and gaps in the existing evidence.

This scoping review was conducted according to the Preferred Reporting Items for Systematic Reviews and Meta-Analysis (PRISMA) 2020 guidelines (Figure 1). The methodological framework for scoping reviews was used, made up of five stages: identifying the research question, identifying relevant and recent studies, selecting studies, charting the data, and summarizing and reporting the findings.

### 2.1. Aims and Eligibility Criteria

This review sought to analyze the existing literature regarding the use of phytotherapeutic, homeopathic agents and bee products in the treatment of acute infections among children aged 0 to 18 years. Eligible studies were original articles published in English or Italian, focusing on acute infections. Included studies comprised randomized controlled trials, cohort studies, and case–control studies. We excluded papers published exclusively as abstract or research letters, studies involving adults, animals (non-human studies), ex vivo or in vitro research, opinion articles, editorials, non-comparative study designs, narrative reviews, and studies on pharmaceutical drugs. Papers addressing conditions without a clear context of infections were also ruled out. Gastrointestinal infections were not considered since the use of non-antibiotic treatments (e.g., probiotics) in these conditions has been already widely addressed [14].

### 2.2. Search Strategy

The following databases: Pubmed/MEDLINE, Embase, Web of Science, and CINAHL. The following terms were used to search the above databases: “infections”, “bacterial”, “viral”, “children”, “pediatric”, “pediatric community”, “pediatric population”, “homeopathy”, “phytotherapy”, “immunostimulants”, “natural remedies”. The search was conducted on 20 December 2024 and the literature search update was conducted on 1 April 2025. Two authors independently performed the initial selection of titles and abstracts, retrieved the full-text articles, and assessed their relevance. Any discrepancies during the full-text screening process were resolved through discussion between the two reviewers and, when necessary, with input from one of the lead investigators. Details about search strategy are in Appendix A.

### 2.3. Data Extraction

Data extraction was carried out using an Excel dataset to collect information from the selected studies. The following data were extracted: name of the first author, year of publication, country, study design, sample size, age, ethnicity, relevant characteristics, type of infection, intervention, dose of intervention, duration, infection course, aims and outcomes, and results (including safety and tolerability). All data were independently extracted by two authors using a standardized form to ensure both consistency and accuracy throughout the process. Disagreements during this process were resolved through discussion or by consulting an additional reviewer.

### 2.4. Inter-Rate Agreement

To ensure consistency and minimize selection bias, two reviewers independently screened all retrieved articles for eligibility. Agreement between reviewers was assessed using Cohen’s kappa (κ) statistic, which measures the level of concordance beyond chance. In cases of disagreement, consensus was reached through discussion or, when necessary, by consulting a third reviewer.

### 2.5. Quality Assessment

The Cochrane Collaboration tool was used to assess the risk of bias (RoB) for RCTs (Figure 2), while the Strobe Statement was applied to evaluate observational studies (Figure 3). Two reviewers independently performed the quality assessment, evaluating the potential for bias in the literature and categorizing studies as “high risk,” “low risk,” or “unclear risk”.

### 2.6. Outcomes

The primary outcomes focus on the reduction in the duration and severity of symptoms, improvement in recovery rates, and decreased use of antibiotics.

The secondary outcomes assess the safety and tolerability of the phytotherapeutic, homeopathic remedies and bee products used for the symptomatic treatment of infections in the pediatric population. These include monitoring for adverse effects, evaluating overall treatment adherence, and comparing tolerability profiles with standard pharmacological therapies.

### 2.7. Data Synthesis and Statistical Analysis

Characteristics of the included studies were reported using descriptive tables. A narrative synthesis of the key findings was performed. We divided the different nutraceuticals into three main groups: phytotherapeutic, homeopathic substances and bee products. For the treatment of URTIs, bee products, such as honey, propolis, and royal jelly, were considered different and specific elements, based on the indications in scientific literature.

## 3. Results

A total of 35 studies were independently screened by the two reviewers, comprising 33 RCTs and 2 observational studies. The reviewers agreed on the inclusion of 31 RCTs and 1 observational study, while discrepancies occurred for 2 RCTs and 1 observational study. The overall agreement between reviewers was high, with a Cohen’s kappa (κ) of 0.74, indicating a substantial level of agreement.

A total of thirty-five studies were identified, including two observational studies [48,49] and thirty-three randomized controlled trials [15,16,17,18,19,20,21,22,23,24,25,26,27,28,29,30,31,32,33,34,35,36,37,38,39,40,41,42,43,44,45,46,47]. According to the World Bank classification [50], the two observational studies were conducted in high income level countries. Of the randomized controlled trials, 18 were conducted in high income level countries [15,16,17,18,19,20,21,22,23,24,25,26,27,28,29,30,48,49], 15 in upper middle-income countries [31,32,33,34,35,36,37,38,39,40,41,42,43,44,45], and two in lower middle-income countries [46,47]. One multicenter study involved both high- and upper-middle income countries (Ukraine and Germany) [44].

### 3.1. Quality Assessment

In the two observational studies [48,49] both were assessed as having a low risk of bias based on title and abstract and on the introduction. However, the evaluation of study methods divided into study design, setting, participants, variables, data sources/measurement, bias, study size, quantitative variables, and statistical methods, revealed some concerns. Notably, one study [49] demonstrated a high risk of bias in the domain of bias, and the other [48] showed a high risk in statistical methods.

The assessment of the results section, which was organized into participants, descriptive data, outcome data, main results, and additional analyses, indicated that one study raised some concerns in other analyses, while the remaining sections showed a low risk of bias.

In terms of discussion, all three studies were assessed as having a low risk of bias in reporting key results. However, interpretation and generalizability showed some concerns. One study presented a high risk of bias in limitation.

Finally, the evaluation of funding sources found a low risk of bias across both observational studies [48,49].

Among the thirty-three randomized controlled trials, 24% (n = 8) were identified as having low overall risk of bias regarding overall risk of bias, 67% (n = 22) of the trials were assessed as having some concerns, and 9% (n = 3) were found to have a high risk of bias [15,21,30].

Regarding the randomization process, three studies [15,21,30] exhibited high risk of bias while eight studies [25,29,31,33,38,40,41,44] showed some concerns. For the domain assessing deviations from intended interventions, nine studies [16,17,21,26,31,37,42,45,46] showed some concerns, while the remainder were assessed as having a low risk of bias.

In terms of missing outcome data, only four studies [24,26,38,43] showed some concerns. For the measurement of the outcome, six studies [19,28,37,38,42] were found to have some concerns. Lastly, in the domain of selection of the reported results, fourteen studies [16,18,19,20,24,27,29,40,41,42,43,45,46,47] exhibited some concerns. Details of the quality assessment are presented in Table 1.

### 3.2. Efficacy

The available evidence suggests that certain complementary approaches might potentially contribute to symptom relief and support recovery in pediatric acute infections, either independently or alongside standard therapies. While some studies have indicated possible improvements in symptom duration and severity, these findings remain inconclusive due to variations in study design and product formulations. Additionally, the safety and tolerability profiles of these interventions appear to be generally favorable when used as adjunctive treatments. However, the heterogeneity of product composition and the methodological limitations of existing studies underscore the necessity for more rigorous and standardized research to clarify their efficacy and ensure their safe integration into pediatric infection treatments. Data on the efficacy of phytotherapeutic, homeopathic remedies and bee products according to the different infectious conditions are provided below.

### 3.3. Upper Respiratory Tract Infections (URTIs)

For URTIs, phytotherapy interventions (e.g., *Pelargonium sidoides*, *Echinacea purpurea*, multi-herb formulae) and bee products were analyzed separately from homeopathic preparations.

Numerous studies [15,16,17,22,25,26,27,30,31,33,36,38,39,40,42,43,47], included in current survey, have evaluated the efficacy of various herbal and homeopathic treatments for URTIs in children. These investigations encompass a range of natural substances, including *Echinacea purpurea* extract, *Pelargonium sidoides*, honey and other bee products, and a range of plant-derived extracts such as gentian root, *vervain herb*, elderflower (*Sambuci flox*), sorrel herb (*Rumex herba*), primrose flowers, *Bambusae textilis McClure*, *Crocus sativus*, *Radix solms-laubachiae*, *Santali albi lignum*, and *Lagotis brevituba Maxim*. These substances were evaluated for their potential in alleviating symptoms and shortening illness duration.

#### 3.3.1. Phytotherapeutic Remedies

##### *Echinacea purpurea* Extract

Three clinical studies [26,27,30] investigated the efficacy of *Echinacea purpurea* in treating URTIs in children, yielding mixed results. Taylor et al. [27] performed an RCT involving 524 children aged 2 to 11 years. Their findings indicated no significant differences between the *Echinacea* and placebo groups concerning the duration or severity of URTI symptoms. Additionally, *Echinacea* use was associated with an increased risk of rash. Conversely, Spasov et al. [26] reported that Kan Jang^®^, a fixed herbal combination, significantly reduced the duration of nasal secretion and congestion (*p* < 0.05) and accelerated recovery rates compared to the *Echinacea* group. Children treated with Kan Jang also exhibited a substantial reduction in the need for additional medications. Weishaupt et al. [30] examined an *Echinacea purpurea* extract in 79 children with cold episodes. The study observed a reduction in the mean number of cold episodes by up to 1.7 days (*p* = 0.02), and a 4–6% decrease in antibiotic prescriptions. Higher doses of the extract led to improvements in symptoms such as runny nose, cough, and sore throat, although these findings were not statistically significant. Regarding school absenteeism, the high-dose group recorded 410 missed school days compared to 494 in the low-dose group.

Overall, the products described were well tolerated, with no serious adverse effects reported.

##### *Pelargonium sidoides* 

Five RCT studies [15,31,33,38,43] assessed the therapeutic efficacy of *Pelargonium sidoides* in children and adolescents with conditions such as the common cold, nasal congestion, and acute tonsillopharyngitis. All studies reported improvements in respiratory symptoms, particularly in alleviating nasal congestion cough, and substantial reduction in the Tonsillitis Severity Score (TSS) [26,27,28]. Specifically, Patiroglu et al. [38] observed an increase in appetite among treated children (*p* = 0.022), while Gökçe et al. [33] noted a statistically notable reduction in cough frequency (*p* = 0.023). In all trials, the treatment was considered effective without adverse effects, suggesting a favorable safety profile for pediatric use.

##### Honey and Bee Products

Seven studies investigated the efficacy of honey and other bee products [16,17,22,25,39,42,47] in treating children, focusing primarily on symptom severity and duration.

Specifically, Cohen et al. [17] and Waris et al. [47] reported improvements in cough severity and frequency, as well as overall health status, including enhanced sleep quality for both children and parents (*p* < 0.014, *p* < 0.018, respectively). In Cohen et al. [16], many children experienced complete resolution of cough symptoms within seven days of initiating supplementation. Peixoto et al. [39] evaluated a honey-bromelain combination in 60 children with irritative cough, comparing it to a placebo group. Both groups showed a reduction in cough episodes within 30 min of administration; however, these results were not statistically significant, aligning with the findings of Nishimura et al. [22].

Seçilmiş et al. [25] demonstrated an influential reduction in Canadian Acute Respiratory Illness and Flu Scale (CARIFS) scores in the treatment group receiving conventional antibiotics combined with various bee products compared to the antibiotic-alone group (*p* < 0.05). Lastly, Shadkam et al. [42] compared the effects of honey with dextromethorphan and diphenhydramine in children aged 24–60 months on their nightly cough. The study found statistically significant improvements in cough frequency and severity in the honey group compared to the control group. The different types of honey and bee products were well tolerated, with no adverse effects reported.

##### Various Medical Plants

Two studies dealt with the effectiveness of medicinal plant combinations [36,41] in reducing the severity and duration of symptoms, as well as improving recovery rates in pediatric patients. Both studies also investigated the potential to reduce the duration of cough and rhinosinusitis-related symptoms and the need for antibiotic prescriptions.

Popovych et al. [41] did not yield statistically significant results but demonstrated good tolerability of the herbal formulation. In contrast, Luo et al. [36] reported that the treatment group experienced a significantly shorter time to cough resolution (*p* = 0.003), with a median of 2 days compared to 3 days in the conventional treatment group (*p* < 0.001). Moreover, the 4-day cough resolution rate was higher in the treatment group (94.4%) than in the control group (74.6%, *p* = 0.001).

#### 3.3.2. Homeopathic Remedies

Seven different studies [19,29,37,40,44,45] investigated the use of homeopathic remedies in the symptomatic treatment of URTIs, focusing on symptoms relief and severity. All studies reported positive outcomes regarding the effectiveness and safety of the treatments. Torbicka et al. [29] assessed a homeopathic combination containing *Vincetoxicum hirundinaria* in patients with RSV infection. They found that the average Symptom Intensity Score (SIS) was lower in the treatment group compared to the control group (3.0 ± 1.6), indicating greater symptom improvement. Malapane et al. [37] administered a homeopathic complex to a pediatric population with acute viral tonsillitis and observed a difference in mean pain ratings between the homeopathic and placebo group (U = 38.000; *p* = 0.001). Additionally, there was a notable decrease in mean tonsil scores in the treatment group compared to the placebo group. Van Haselen, Jacobs, and Voss et al. [19,44,45] observed reductions in symptom duration and significant improvements in symptoms such as runny nose, fever, and malaise. They also noted decreases in scores such as the Wisconsin Upper Respiratory Symptom Survey (WURSS-21) and CARIFS. Popovych et al. [40] treated a group of 238 children with BNO 1030 extract and showed improvements in various symptoms from day 1 to day 4, along with a reduction in antipyretic use.

### 3.4. Bronchitis

#### Phytotherapeutic Remedies

Evidence for phytotherapy (especially *Pelargonium sidoides* EPs 7630 and multi-herb formulas) in acute bronchitis in children is more developed, with several RCTs showing reductions in Bronchitis Severity Scale (BSS) and faster symptom resolution. No eligible homeopathy trials for acute lower respiratory tract infections were identified.

Several studies have investigated the efficacy of herbal and homeopathic treatments for pediatric bronchitis, focusing on symptom severity, particularly cough, and overall infection duration. Notably, research on herbal compounds has yielded significant findings, while evidence supporting homeopathic remedies remains limited.

*Pelargonium sidoides*, particularly the EPs 7630 extract, has been extensively studied for its effectiveness in treating acute bronchitis in children [20,32,34,48,49].

Haidvogl and Kamin et al. [20,48] found a significant reduction in the BSS, from 6-0 ± 3.0 at baseline to 2.7 ± 2.5 after one week and 1.4 ± 2.1 by the end of the study (*p* < 0.001). Improvements were most pronounced for symptoms such as cough and pulmonary rales upon auscultation. The onset of therapeutic effects was rapid, with significant improvements observed as early as day 1–2 and 3–4 (*p* < 0.0001). After 7 days of administration, the EPs 7630 exhibited a greater reduction in BSS compared to the placebo group, indicating both enhanced efficacy and a faster onset of symptom relief. Similarly, Chen et al. [32] reported that children treated with a multi-herbal extract formulation experienced a significant reduction in VAS scores after 7 days of treatment compared to controls (6.35 ± 3.45 vs. 3.73 ± 3.98; *p* < 0.001).

### 3.5. Otitis

#### 3.5.1. Phytotherapeutic Remedies

Articles have been identified that focus solely on the use of homeopathic remedies in the symptomatic treatment of otitis [18,23,24,28,46], while there is a lack of material concerning phytotherapeutics such as *Echinacea*, *Pelargonium sidoides*, and other bee products.

#### 3.5.2. Homeopathic Remedies

While herbal treatments have demonstrated efficacy in managing bronchitis, evidence supporting the use of homeopathic remedies for otitis media is limited. Several studies have explored the effects of homeopathic compounds such as *Chamomilla*, *Belladonna*, *Agraphis nutans*, *Thuya occidentalis*, and *Kalium muriaticum*, in pediatric patients with Acute Otitis Media (AOM) or otitis media with effusion [18,23,24,28,46]. However, only a few studies have assessed the relationship between these treatments and pain relief.

In a randomized, double-blind, placebo-controlled pilot study, Jacobs et al. [18] administered individualized homeopathic remedies to 75 children aged 18 months to 6 years with AOM. The study found a significant decrease in symptom scores within 24 to 64 h following treatment, indicating a positive effect of homeopathy on symptom relief. However, the study also noted that the differences between the homeopathic and placebo groups were not statistically significant over the longer term.

Conversely, Sinha [46] conducted a trial comparing homeopathic treatment with conventional therapy in 81 children. The results showed limited efficacy of homeopathic treatment: by day 7, only 7.58% of children in the treatment group had recovered, compared to 53% in the conventional group (*p* = 0.356); by day 10, recovery was observed in 10.9% of the homeopathic group versus 100% in the conventional group.

Pedrero-Escalas [23] evaluated the impact of homeopathic treatment on symptom relief in children. In the experimental group, 61.9% of children were cured compared to 56.8% in the placebo group, with no significant difference. Adverse events are similar, except for fewer URTIs in the first group (3 vs. 13, *p* = 0.001). Overall, homeopathic treatment showed no significant efficacy. Therefore, it cannot be considered an effective treatment for children with AOM.

### 3.6. Hand-Foot-Mouth Disease (HFMD)

#### Phytotherapeutic Remedies

One phytotherapy RCT showed modest benefits; no homeopathy trials were found.

Regarding HFMD, a viral illness, the literature offers limited evidence. One of the most relevant studies is by Liu et al. [35], which evaluated the efficacy of Jinzhen oral liquid, a formulation containing *Salgae tataricae cornu*, *Fritillaria usuriensis maxim*, *Scutellaria baicalensis georgi*, and other herbal extracts, in a pediatric population of 399 children aged 1–7 years, over a 7-day period. The study assessed outcomes such as time to the first disappearance of oral ulcers and hand/foot vesicles and time to fever clearance. Results showed that children treated with Jinzhen experienced a significantly shorter time to symptom resolution compared to placebo (4.9 vs. 5.7 days; *p* = 0.0036), faster fever reduction (43.42 h vs. 54.92 h; *p* = 0.0161), and a 28.5% lower risk of persistent symptoms (*p* = 0.0032).

### 3.7. Flu

#### Phytotherapeutic Remedies

One phytotherapy study suggested shorter fever duration; no homeopathy trials were found.

The only available data concerning the treatment of flu A with herbal medicine, specifically Mao-to, a traditional Japanese herbal remedy, comes from a study by Kubo [21]. This study observed the effectiveness of phytotherapeutic treatment in reducing the duration of symptoms, particularly fever. In the study, 24 children in the treated group were administered only Mao-to for two days, and the average duration of fever was found to be significantly shorter in the treatment group compared to the control group (*p* < 0.01). The treatment appears to be safe, as no side effects were reported.

## 4. Discussion

### 4.1. Efficacy of Phytotherapeutic, Homeopathic Interventions and Bee Products in Pediatric Acute Infections

This scoping review synthesized data from 35 studies, mostly RCTs and double-blind trials, with only 2 observational studies of acceptable quality. While this rigor supports reliability, variability in protocols, dosages, and outcomes measures introduces marked heterogeneity, limiting direct comparisons and precluding meta-analysis. Some phytotherapeutic agents improved symptoms in well-defined pediatric infections such as URTIs, bronchitis, and otitis media. These results, however, should be interpreted cautiously given heterogeneity and small sample sizes.

The main findings can be summarized as follows: (i) certain herbal treatments, such as *Pelargonium sidoides* and bee products like honey showed moderate benefits in respiratory acute infections, while evidence for homeopathic remedies was very limited, inconsistent, and often low-quality (ii) phytotherapeutics, homeopathics and bee products appeared generally safe, though safety data remain sparse for some interventions (iii) important gaps include the absence of data on non-respiratory infections such as urinary tract, dermatological, central nervous system, and osteoarticular infections. Even among higher-quality studies, heterogeneity in design and outcomes limits generalizability of the findings. Collectively, these observations highlight the need for more rigorous research to clarify the efficacy and safety of phytotherapeutic, homeopathic remedies and bee products across a broader range of infections.

### 4.2. Phytotherapeutic Treatments and Bee Products

Phytotherapeutics, especially *Pelargonium sidoides*, *Echinacea purpurea*, and various plant extracts and bee products like honey have been tested in pediatric URTIs, bronchitis, HFMD, and flu. *Pelargonium sidoides* improved cough and overall health in acute bronchitis, though evidence is limited by small samples and variable outcomes. Proposed mechanisms include antimicrobial, immunomodulatory, and mucociliary effects [51]. In vitro, *Pelargonium sidoides* inhibited Gram-positive and negative bacteria, showed antifungal activity, and prevented viral replication. It also stimulated cytokine and interferon release, enhanced phagocytosis, and increased ciliary beat frequency, improving mucus clearance [52].

*Echinacea purpurea* is traditionally used for colds and respiratory infections. It may stimulate macrophages, natural killer, and cytokine production, with polysaccharides and alkamides as active compounds. These immunomodulatory effects might help the body better respond to infections, particularly respiratory tract infections [53].

Honey exhibits antimicrobial and wound-healing properties [54], through osmolarity, low pH, hydrogen peroxide, and antioxidants and bioactive compounds [55]. Its viscous texture also creates a protective barrier, aiding wound healing and soothing irritated mucous membranes [56]. Overall, the findings are broadly consistent with prior literature but should be interpreted with caution [57,58].

### 4.3. Homeopathic Treatments in Pediatric Acute Infections

Homeopathic remedies studied in pediatric URTIs and otitis media include *Vincetoxicum hirundinaria*, BNO 1030 extract, and combinations of *Chamomilla*, *Belladonna*, *Agraphis nutans*, *Thuya occidentalis*, and *Kalium muriaticum*. Some RCTs [19,29,37,40,44,45,48] reported reductions in symptom severity, especially cough and reduced need for conventional antipyretics, but effects were often not statistically significant or based on small samples. Similarly, few studies [18,28,46] suggested improvements in otalgia, transient hearing loss, and school absenteeism. However, none of the studies mentioned were of high quality, and the only rigorous trial showed no efficacy in otitis media with effusion [23]. Overall, evidence for homeopathy remains weak and inconclusive.

### 4.4. Gaps in the Current Literature and Future Research Perspectives

This review found in phytotherapeutic, homeopathic and bee products studies for pediatric infections beyond URTIs, such as urinary or infections. Preventive uses were excluded due to lack of data. Research focuses almost exclusively on bacterial and viral infections, with no trial on fungal or other pathogens. Conditions like HFMD and flu remain underexplored. Age distribution also limits generalizability: phytotherapeutic treatments were primarily tested in children 1–14 years, while, homeopathy included neonates and infants under 1 year [23,28,29,45]. This variability complicates comparisons. Standardization of age groups would improve study comparability. Emerging evidence suggests potential benefits of probiotics in URTIs. Future research could explore combined interventions, such as phytotherapeutics with other functional compounds (e.g., probiotics and/or probiotics). Table 2 summarizes the current research gaps and potential directions for future investigation.

### 4.5. Limitations of the Scoping Review

This scoping review has several limitations that warrant consideration. The included studies exhibited significant heterogeneity in methodologies, dosages, type of interventions, category of actives, populations, and outcomes, which may affect the generalizability of our findings. Additionally, some data was self-reported, introducing potential biases such as recall or social desirability bias. Socioeconomic disparities among the study population were also evident, which could influence the outcomes and their applicability to different demographic groups. Furthermore, four studies were industry sponsored, raising questions about potential conflicts of interest and the risk of reporting bias. These factors collectively underscore the need for cautious interpretation of our findings and highlight areas for improvement in future research as previously detailed.

## 5. Conclusions

Phytotherapeutic agents such as *Pelargonium sidoides* and bee products like honey have shown some evidence of benefit in reducing symptom severity and duration in children and adolescents (1–18 years) with URTIs. These interventions were generally well tolerated, with only mild adverse effects reported. In contrast, the evidence for homeopathic treatments is weak, inconsistent, and of insufficient quality to support their recommendation in clinical practice. Overall, the evidence based on these approaches is heterogeneous, limited in scope, and affected by methodological variability. Consequently, while certain phytotherapeutic agents may be considered as complementary options in pediatric acute respiratory infections, these findings cannot be extrapolated to other types of infections. Taken together, phytotherapy and bee products may be considered as a complementary option for selected pediatric respiratory infections, while homeopathic treatments lack consistent, high-quality evidence of efficacy; these remedies require further rigorous evaluation.

## Figures and Tables

**Figure 1 nutrients-17-03137-f001:**
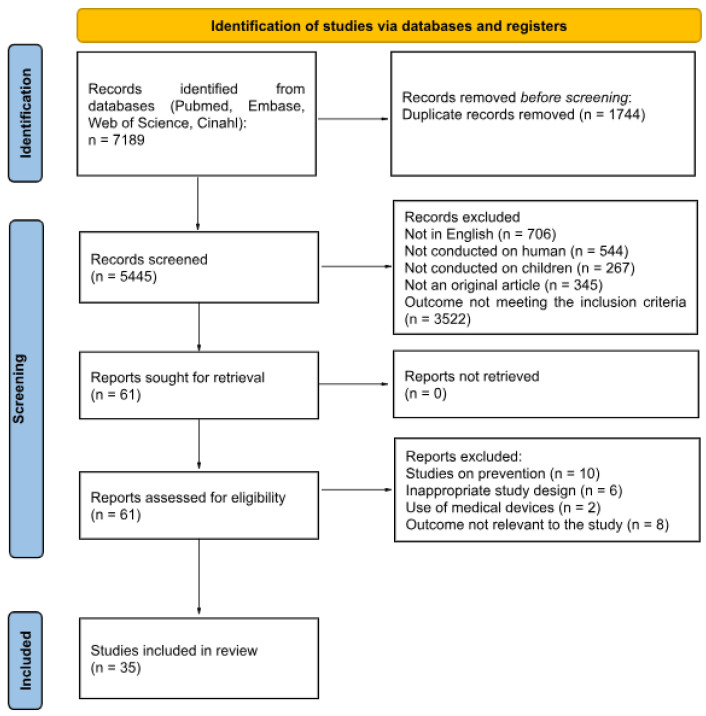
Literature review process.

**Figure 2 nutrients-17-03137-f002:**
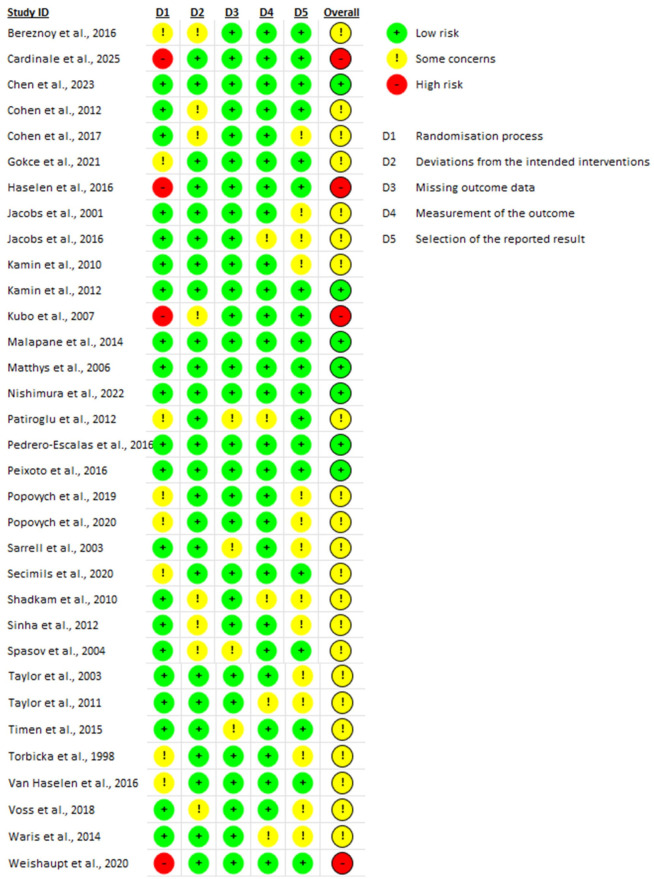
Quality assessment (RCTs) [15,16,17,18,19,20,21,22,23,24,25,26,27,28,29,30,31,32,33,34,35,36,37,38,39,40,41,42,43,44,45,46,47].

**Figure 3 nutrients-17-03137-f003:**
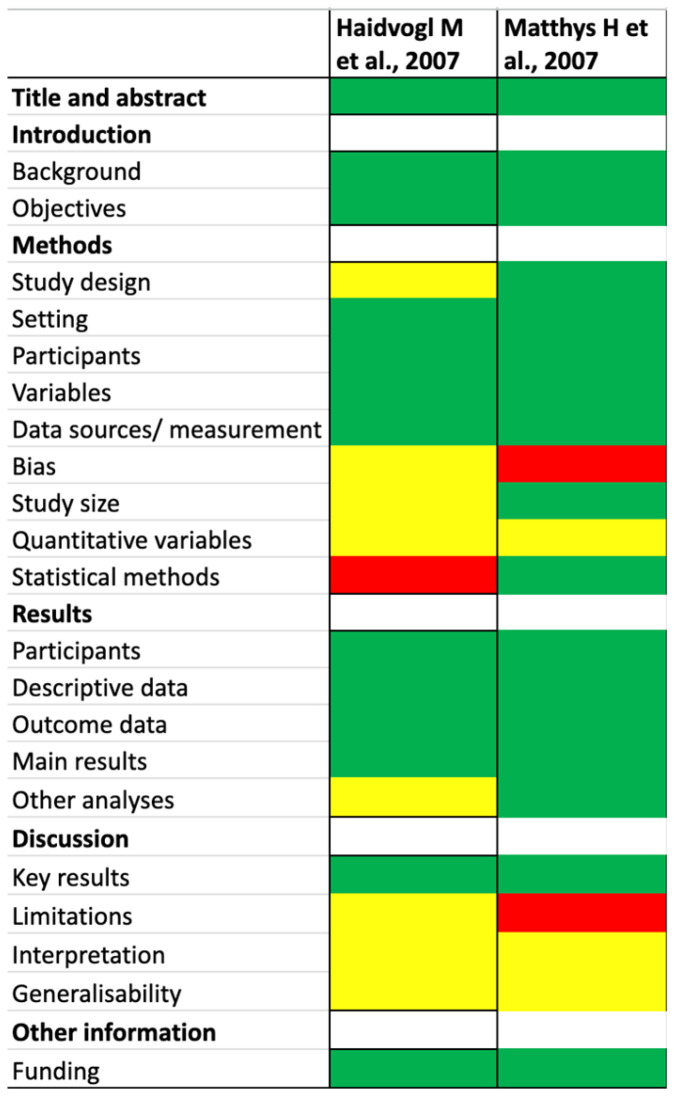
Quality assessment (observational studies) [48,49]. The different colors (red, yellow, and green) refer to the clinical trial risks.

**Table 1 nutrients-17-03137-t001:** RCTs and observational studies.

Author	Country	Income	Study Design	Population Size	Age	Infection	Category of Dietary Supplementation	Actives	Intervention vs. Placebo or Standard Therapy
**Bereznoy VV et al., 2016** [31]	Ukraine	Upper-middle income	RCT	126	6–10 years	URTI	Phytotherapy	*Pelargonium sidoides*	Treatment vs. Placebo
**Cardinale F et al., 2025** [15]	Romania	High income	RCT	130	3–10 years	URTI	Phytotherapy and bee products	*Pelargonium sidoides*, honey, propolis and zinc	Treatment + standard therapy vs. Standard therapy
**Chen HF et al., 2023** [32]	China	Upper-middle income	RCT	443	1–14 years	Bronchitis	Phytotherapy	Arctium Lappa, Morus Alba, Mentha Haplo Calyx, Sabillina Tenuifolia, Fritillaria Unibracteata, Peucedanum Praeruptorum Dunn, Aster Tataricus, Citrus Aurantium, Isatis Indogotica, Glycyrrhiza Uralensis.	Treatment vs. 10% treatment
**Cohen HA et al., 2017** [16]	Israel	High income	RCT	141	2–5 years	URTI	Phytotherapy and bee products	Polysaccharides, Resin, Honey	Treatment vs. standard
**Cohen HA et al., 2012** [17]	Israel	High income	RCT	270	1–5 years	URTI	Phytotherapy and bee products	Eucalyptus honey, citrus honey, labiatae honey	Treatment 1 or treatment 2 or treatment 3 vs. placebo
**Gökçe Ş et al., 2021** [33]	Turkey	Upper-middle income	RCT	164	1–18 years	URTI	Phytotherapy	*Pelargonium sidoides*	Treatment vs. placebo
**Haidvogl M et al., 2007** [48]	Germany	High income	Observational study	742	1–12 years	Bronchitis	Phytotherapy	*Pelargonium sidoides*	x
**Jacobs J et al., 2001** [18]	United States	High income	RCT	75	18 months–6 years	Acute otitis media	Homeopathy	Pulsatilla Ngrans, *Chamomilla*, Sulfur, Calcarea Carbonica	Treatment vs. placebo
**Jacobs J et al., 2016** [19]	United States	High income	RCT	261	2–5 years	URTI	Homeopathy	Allium Cepa, Hepar Sulphuris Calcareum, Natrum Muriaticum, Pulsatilla, Hydrastis, Extract of Glycyrrhiza	Treatment vs. placebo
**Kamin W et al., 2010** [34]	Ukraine	Upper-middle income	RCT	200	1–18 years	Bronchitis	Phytotherapy	*Pelargonium sidoides*	Treatment vs. placebo
**Kamin W et al., 2012** [20]	Russia	High income	RCT	220	1–18 years	Bronchitis	Phytotherapy	*Pelargonium sidoides*	Treatment vs. placebo
**Kubo T et al., 2007** [21]	Japan	High income	RCT	49	5 months–13 years	Flu A	Phytotherapy	Ephedrae herba, Armenicae cortex, Cinnamon cortex, Glycyrrhizae radix	Anti-viral therapy + treatment or treatment or anti-viral treatment
**Liu J et al., 2014** [35]	China	Upper-middle income	RCT	399	1–7 years	HFMD	Phytotherapy	*Salgae tataricae cornu*, *Fritillaria usuriensis maxim*, Rheum Officiale Baill, *Scutellaria Baicalensis Georgi*, Sulfate minerals gypsum, *Glycyrrhiza glabra* L.	Treatment vs. placebo
**Luo H et al., 2022** [36]	China	Upper-middle income	RCT	138	2–14 years	URTI	Phytotherapy	*Bambusae textilis McClure*, *Crocus sativus*, Bovis calculus artifactus, Bergeniae rhizome, Aconitum tanguticum, Glycyrrhizae radix et rhizome, *Radix solms-laubachiae*, *Lagotis brevituba maxim* and *Santali albi lignum*	Treatment vs. standard therapy
**Malapane E et al., 2014** [37]	South Africa	Upper-middle income	RCT	30	6–12 years	URTI	Homeopathy	*Atropa belladonna*, Calcarea Phosphoricum, Hepar Sulphuris, Kalium Bichromat, Mercurius protoiodid, and Mercurius biniodid *Kalium muriaticum*	Treatment vs. placebo
**Matthys H et al., 2007** [49]	Germany	High income	Observational study	2099	Children and adults aged 0–93 years	Bronchitis	Phytotherapy	*Pelargonium sidoides*	
**Nishimura T et al., 2022** [22]	Japan	High income	RCT	161	1–5 years	URTI	Bee products	Acacia honey	Treatment vs. placebo
**Patiroglu T et al., 2012** [38]	Turkey	Upper-middle income	RCT	28	1–5 years	URTI	Phytotherapy	*Pelargonium sidoides*	Treatment vs. placebo
**Pedrero-Escalas MF, 2016** [23]	Spain	High income	RCT	96	2 months–12 years	Otitis media	Homeopathy	*Agraphis nutans*, *Thuya occidentalis*, *Kalium muriaticum* and Arsenicum Iodatum	Treatment vs. placebo
**Peixoto Dm et al., 2016** [39]	Brazil	Upper-middle income	RCT	60	2–15 years	URTI	Phytotherapy and bee products	Bromelin + honey	Treatment (bromelin + honey vs. placebo (honey)
**Popovych V et al., 2019** [40]	Ukraine	Upper-middle income	RCT	238	6–18 years	URTI	Homeopathy	BNO 1030 extract	Treatment + standard therapy vs. standard therapy
**Popovych V et al., 2020** [41]	Ukraine	Upper-middle income	RCT	292	6–11 years	URTI	Phytotherapy	Gentian root, Vervain herb, *Sambuci flox*, *Rumex herba*, primrose flowers	Treatment + standard therapy vs. standard therapy
**Sarrell EM et al., 2003** [24]	Israel	High income	RCT	171	5–18 years	Otitis media	Homeopathy	Allium sativum, verbascum thapsus, calendula flores, hypercium perfoliatum, lavender, vitamin E	Treatment or treatment + topical anesthetic drops or treatment + antibiotic, or antibiotic + anesthetic drops.
**Seçilmiş Y et al., 2020** [25]	Canada	High income	RCT	104	5–12 years	URTI	Bee products	Honey, royal jelly, propolis	Bacterial infection: treatment + antibiotic vs. antibiotics Viral infection: Treatment vs. placebo
**Shadkam MN et al., 2010** [42]	Iran	Upper-middle income	RCT	139	2–5 years	URTI	Bee products	Honey	Treatment or standard therapy A or standard therapy B or control group
**Sinha MN et al., 2012** [46]	India	Lower-middle income	RCT	81	2–6 years	Otitis media	Homeopathy	*Belladonna*, Hepar sulphuris, Calcarea carbonica, *Chamomilla*, Mercurius solubilis	Treatment vs. standard therapy
**Spasov AA et al., 2004** [26]	Russia	High income	RCT	133	4–11 years	URTI	Phytotherapy	Kan Jang or *Echinacea purpurea*	Treatment A + standard therapy or treatment B + standard therapy or standard therapy
**Taylor JA et al., 2003** [27]	United States	High income	RCT	524	2–11 years	URTI	Phytotherapy	*Echinacea purpurea*	Treatment vs. placebo
**Taylor J et al., 2011** [28]	United States	High income	RCT	120	6 months–11 years	Otitis media	Homeopathy	Pulsatilla, *Chamomilla*, *Belladonna*, lycopodium	Treatment + standard therapy or standard therapy
**Timen G et al., 2015** [43]	Ukraine	Upper-middle income	RCT	78	6–10 years	URTI	Phytotherapy	*Pelargonium sidoides*	Treatment vs. placebo
**Torbicka E et al., 1998** [29]	Poland	High income	RCT	128	Infants	URTI	Homeopathy	*Vincetoxicum hirundinaria*, sulfur	Treatment + standard therapy vs. standard therapy
**Van Haselen R et al., 2016** [44]	Ukraine and Germany	Upper-middle income and high income	RCT	261	1–11 years	URTI	Homeopathy	Aconitum, Bryonia, Eupatorium Perfolatum, Gelsemium, Ipecuanha, Phosphorus	Treatment + standard therapy vs. standard therapy
**Voss HW et al., 2018** [45]	Ukraine	Upper-middle income	RCT	180	7 months–12 years	URTI	Homeopathy	Drosera, Coccus cacti, Cuprum Sulfuricum, Ipecacuanha	Treatment vs. placebo
**Waris A et al., 2014** [47]	Kenya	Lower-middle income	RCT	133	1–12 years	URTI	Bee products	Honey	Treatment vs. standard therapy vs. placebo
**Weishaupt R et al., 2020** [30]	Switzerland	High income	RCT	79	4–12 years	URTI	Phytotherapy	Exctract of *Echinacea purpurea*	5 tablets of treatment or 3 tablets of treatment

**Table 2 nutrients-17-03137-t002:** Clinical research and possible future studies on phytotherapeutic and homeopathic remedies in pediatric acute respiratory conditions.

Current Areas of Clinical Research	Possible Future Studies
Use of homeopathic remedies in pediatric respiratory infections by high quality studies	Testing homeopathic treatments in acute respiratory infections like flu, bronchitis, and otitis media in children using well powered blinded RCT placebo vs. interventions and employing validated scores for outcomes assessment
Use of phytotherapeutic agents (e.g., *Pelargonium sidoides*) in school-age children	Testing phytotherapeutics in infants and neonates under 1 year of age
Isolated testing of phytotherapeutic treatments or phytotherapeutics and other approaches	Studying combined use of more phytotherapeutics or phytotherapeutics with other potentially synergic compounds (e.g., probiotics)
Single-agent studies of *Pelargonium sidoides* or honey	Evaluating synergic combinations such as *Pelargonium sidoides* + honey or *Pelargonium sidoides* + probiotics
Effectiveness and safety pf phytotherapeutic agents in young children	Testing phytotherapeutic in infants and children <1 year of age

## Data Availability

The original contributions presented in the study are included in the article, further inquiries can be directed to the corresponding authors.

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
