# Peer review of "Phytotherapeutic, Homeopathic Interventions and Bee Products for Pediatric Infections: A Scoping Review"

_nutrients, 2025, doi:10.3390/nu17193137_

Round 1

Reviewer 1 Report

Comments and Suggestions for Authors

Manuscript: "Phytotherapic and Homeopathic Interventions for Pediatric Infections: A Scoping Review"

Manuscript ID:  nutrients-3855498

#  Global evaluation

This scoping review explores the role of phytotherapeutic and homeopathic interventions in pediatric infections. This is a timely and relevant topic, given the growing interest in complementary and alternative medicine (CAM) for children. Safety, efficacy and integration with conventional medicine are all crucial concerns in this area. The review adds value by synthesizing scattered literature and attempting to provide a global perspective on therapies that are often culturally contextualised.

The manuscript's strength lies in its subject matter, its use of the scoping review methodology, and its relevance to pediatric healthcare, integrative medicine, and public health. However, the text requires refinement to improve conceptual precision, clarity of scope, methodological transparency and critical depth. Some sections read more like a narrative overview than a systematic scoping review, and the conclusions sometimes go beyond the evidence presented.

# Conceptual Issues

-Scope Ambiguity. The manuscript does not clearly define what is included under “phytotherapic” versus “homeopathic,” leading to possible conceptual overlap or misclassification.

-Evidence Hierarchy. Scoping reviews should not assess efficacy per se but map existing evidence. At times, the manuscript seems to conflate descriptive mapping with evaluative claims.

-Pediatric Focus. While pediatric infections are central, some examples lack child-specific data and rely on general CAM studies.

-Balance of Perspectives. The discussion tends to emphasise positive findings without consistently highlighting methodological weaknesses, risk of bias, or safety concerns (which are essential in pediatric settings).

# Issues by lines

-Abstract. Lacks a structured format (Background, Methods, Results, Conclusion) recommended for scoping/systematic reviews. Overstates “effectiveness” rather than “reported outcomes.”

-Introduction. Needs a clearer rationale. Why a scoping review? What gap in literature does it fill?

-Methods Section. Search strategy, inclusion/exclusion criteria, and data charting process are not described in sufficient detail for reproducibility.

-Results Section. Summaries of included studies are too general; table (s) of study characteristics (population, intervention, outcomes) would add transparency.

-Discussion. At times reads as advocacy rather than critical analysis. Phrases like “proved effective” should be replaced by “reported beneficial outcomes” unless supported by strong evidence.

-Conclusion. Too definitive. Should emphasise that the evidence base is heterogeneous, limited, and requires further high-quality research.

# Suggestions for Improvement

-Clarify the scope and definitions of phytotherapic vs. homeopathic interventions at the outset.

-Strengthen methodology reporting (databases, keywords, screening process, PRISMA-ScR flow diagram).

-Provide structured tables summarising included studies, with clear pediatric relevance.

-Reframe language to remain descriptive and cautious, avoiding causal or efficacy claims.

-Expand critical appraisal of limitations in included studies (e.g., small sample sizes, lack of blinding, publication bias).

-Balance perspectives by equally highlighting safety concerns, adverse effects, and regulatory considerations.

-Revise conclusion to emphasize gaps in evidence and the need for more robust pediatric-focused research.

Author Response

REVIEWER 1

#  Global evaluation

This scoping review explores the role of phytotherapeutic and homeopathic interventions in pediatric infections. This is a timely and relevant topic, given the growing interest in complementary and alternative medicine (CAM) for children. Safety, efficacy and integration with conventional medicine are all crucial concerns in this area. The review adds value by synthesizing scattered literature and attempting to provide a global perspective on therapies that are often culturally contextualized.

The manuscript's strength lies in its subject matter, its use of the scoping review methodology, and its relevance to pediatric healthcare, integrative medicine, and public health. However, the text requires refinement to improve conceptual precision, clarity of scope, methodological transparency and critical depth. Some sections read more like a narrative overview than a systematic scoping review, and the conclusions sometimes go beyond the evidence presented.

Authors answer:  Thank you for your comments and the opportunity to improve this manuscript. We hope that the corrections made will make the manuscript more understandable and allow it to be published as soon as possible.

# Conceptual Issues

-Scope Ambiguity. The manuscript does not clearly define what is included under “phytotherapic” versus “homeopathic,” leading to possible conceptual overlap or misclassification.

Authors answer:  Thank you for your comment. At L59 and at L62, we explained the definition of “phytotherapy” and “homeopathy”, according to the European definition. Also, we discussed the properties of these substances (e.g. immunomodulatory, anti-inflammatory, antimicrobial).

-Evidence Hierarchy. Scoping reviews should not assess efficacy per se but map existing evidence. At times, the manuscript seems to conflate descriptive mapping with evaluative claims.

Authors answer:   Thank you for your comment. At L83, we discussed our paper aims. The scoping review aims to synthesize the current literature on the use of phytotherapeutic and homeopathic remedies in managing acute infections in children. Then, we evaluated the state of the art on the interventions in pediatric populations. The discussion about efficacy, safety and mechanisms of action is relevant to identify gaps in literature and propose directions for future research on the role of nutraceuticals in pediatric acute infections. On the other hand, we understand the reviewer’s concern and we clarified this point in the text in the description of the study aims

-Pediatric Focus. While pediatric infections are central, some examples lack child-specific data and rely on general CAM studies.

Authors answer:  Thank you for your comment. We better clarified in the text this issue. In particular, at L127, there is the paragraph about aims and inclusion criteria, in which we point out the inclusion criteria (use of phytotherapeutic and homeopathic agents in the treatment of acute infections among children aged 0 to 18 years. Also, in the supplementary materials (Table A1) we added a specific table with all studies regarding children population. Studies regarding adults are excluded.

-Balance of Perspectives. The discussion tends to emphasise positive findings without consistently highlighting methodological weaknesses, risk of bias, or safety concerns (which are essential in pediatric settings).

Authors answer:  Thank you for your comment. In our discussion (L529) we introduced a specific paragraph about “limitations of the scoping review” in which we discussed about:

- heterogeneity of the studies, population and outcomes

- data self-reported

- socioeconomic disparities

- studies industry sponsored

# Issues by lines

-Abstract. Lacks a structured format (Background, Methods, Results, Conclusion) recommended for scoping/systematic reviews. Overstates “effectiveness” rather than “reported outcomes.”

Authors answer:  Thank you for your comment. Our abstract (L18-L36) is made up of subsections, as recommended for scoping reviews: background, methods, effectiveness, and conclusions. L27: done (results à effectiveness).

-Introduction. Needs a clearer rationale. Why a scoping review? What gap in literature does it fill?: Authors answer: Thank you for your comment. We decided to opt for a scoping review because, as stated in the introduction (L83-91) our goal is to synthesize the current scientific literature on the use of complementary medicine, aiming to fill existing gaps and, where possible, guide future research on the rational use of these substances in the symptomatic treatment of acute infections in pediatric patients. Unlike the systematic review we initially planned, our purpose is not to find a definitive answer, but rather to map the current state of the art. We would like to fill the gaps in the literature about the efficacy and the rationale use of these nutraceutical substances in acute infections in children (L88-89). This point was better clarified in the revised manuscript.

-Methods Section. Search strategy, inclusion/exclusion criteria, and data charting process are not described in sufficient detail for reproducibility

Authors answer:   thank you for your comment. In Supplementary Materials, we added details about literature search of databases (PubMed, Embase, Web Of Science, CINHAL), and inclusion/exclusion criteria (L129-136).

-Results Section. Summaries of included studies are too general; table (s) of study characteristics (population, intervention, outcomes) would add transparency

Authors answer:   thank you for your comment. We modified and added the table with all details about studies in the text; then, we removed “Appendix 1” about Supplementary Material. in Table 2, we included all study characteristics.

-Discussion. At times reads as advocacy rather than critical analysis. Phrases like “proved effective” should be replaced by “reported beneficial outcomes” unless supported by strong evidence

Authors answer:   Thank you for your comment. L416-L571: We have revised the discussions to make them more critical.

-Conclusion. Too definitive. Should emphasise that the evidence base is heterogeneous, limited, and requires further high-quality research

Authors answer:  Thank you for your comment.  L573-L589. We have revised the conclusions to emphasize the aspects underlined.

# Suggestions for Improvement

-Clarify the scope and definitions of phytotherapic vs. homeopathic interventions at the outset.

-Strengthen methodology reporting (databases, keywords, screening process, PRISMA-ScR flow diagram).

-Provide structured tables summarising included studies, with clear pediatric relevance.

-Reframe language to remain descriptive and cautious, avoiding causal or efficacy claims.

-Expand critical appraisal of limitations in included studies (e.g., small sample sizes, lack of blinding, publication bias).

-Balance perspectives by equally highlighting safety concerns, adverse effects, and regulatory considerations.

-Revise conclusion to emphasize gaps in evidence and the need for more robust pediatric-focused research.

Authors answer:  Thank you for raising these points. We think that the manuscript has markedly improved addressing these concerns. We are available to address further points if need.

Reviewer 2 Report

Comments and Suggestions for Authors

This manuscript addresses an important and timely question: the role of phytotherapeutic and homeopathic remedies in the management of pediatric acute infections. The topic is clinically relevant given concerns about antibiotic overuse, parental interest in natural therapies, and the need for evidence-based alternatives. The review is comprehensive, includes a broad literature search, and provides a structured synthesis of existing studies. However, there are several areas where the manuscript could be improved.

1] Although the study is framed as a scoping review, the methodology includes elements more typical of a systematic review (e.g., detailed quality assessment of included studies). Please clarify your rationale and ensure that the description and presentation fully align with scoping review standards (PRISMA-ScR).

2] Please use “phytotherapeutic” consistently rather than “phytotherapic.”

3] Email typo: “unimit.i” instead of “unimi.it” in corresponding author.

4] Some references lack uniform formatting.

Author Response

REVIEWER 2:

This manuscript addresses an important and timely question: the role of phytotherapeutic and homeopathic remedies in the management of pediatric acute infections. The topic is clinically relevant given concerns about antibiotic overuse, parental interest in natural therapies, and the need for evidence-based alternatives. The review is comprehensive, includes a broad literature search, and provides a structured synthesis of existing studies. However, there are several areas where the manuscript could be improved

Authors answer:  Thank you for your comments and the opportunity to improve this manuscript. We hope that the corrections made will make the manuscript more understandable and allow it to be published as soon as possible.

1] Although the study is framed as a scoping review, the methodology includes elements more typical of a systematic review (e.g., detailed quality assessment of included studies). Please clarify your rationale and ensure that the description and presentation fully align with scoping review standards (PRISMA-ScR)

Authors answer:  thank you for your comment. As indicated starting from L93, the study was initially presented as a systematic review, but, due to the heterogeneity of the data and the lack of necessary information in some studies, it was redirected towards a scoping review. The aims were not to demonstrate the efficacy and safety of the investigated substances, but rather to map the overall situation regarding the administration of phytotherapeutic and homeopathic products in pediatric patients with acute infections. Regarding the development of the flowchart, the PRISMA guidelines and the entire quality assessment were nevertheless included in the manuscript, in order to provide greater scientific values and credibility. These additions also highlight that the selected studies are of good quality, according to objective criteria (Strobe and RoB scales).

2] Please use “phytotherapeutic” consistently rather than “phytotherapic”

Authors answer:   thank you for your comment. We corrected the manuscript title (phytotherapeutic and homeopathic interventions) and also we modified L506.

3] Email typo: “unimit.i” instead of “unimi.it” in corresponding author.

Authors answer:  thank you for your comment that was addressed in the revised manuscript.

4] Some references lack uniform formatting.

Authors answer:  thank you for your comment that was addressed in the revised manuscript.

Reviewer 3 Report

Comments and Suggestions for Authors

nutrients-3855498-peer-review-v1

The current review is interesting and gives some information about homeopathic and possible control in human (and why not veterinary) medicine. In my opinion paper can be suggested to the Editor to be considered for publication, however, authors will need to upgrade some parts, adjust other and maybe provide some additional information.

Some specific comments:

I am not sure if the format is really appropriate and it is according to the recommendations form the Journal and Publisher. On my knowledge Foods/Nutrients do not require splitting the abstract in subsections. Please, the technical team and the Editor of the Journal, can you please, give your opinion on this issue?

It is not clear, paper coming as submitted to Nutrients, however, using template of Foods?

I am not sure if the format is really appropriate and it is according to the recommendations form the Journal and Publisher. Please, the technical team and the Editor of the Journal, can you please, give your opinion on this issue?

Ln36-37: Do not need quotation marks for the keywords.

Ln58: In this context, by national, do you mind Italian? Please, specify. In fact, provided definition is Italian or EU agency? Please, specify.

Ln63: Please, use opening citations marks for pathos. Needs to be "pathos".

Section 91-97, it is not really needed to justify what original idea was and why authors have different types of paper builders.

Approaches for the screening for appropriate literature and analysis of the recorded informative sources is well described and presenting authors strategy in preparation of current manuscript.

In my opinion authors describing with a bit over detail how selected studies were choice to be included in analysis, why is important to look not only at title and abstract, but that is also a kind of obvious criterium to read entire paper if you will use it as reference (however, some colleagues think that only checking abstract is sufficient, this is philosophical comment, nothing with the authors of the current paper). In my opinion, this part of the manuscript may be a bit reduced and later given a bit more focus on the examples of different homeopathic products and their role.

Ln232: If I am not wrong, abbreviation UTTIs is for the first time used in this place. Please, in this and similar cases use full name before you introduce abbreviation, even for the well-known abbreviations. Moreover, maybe it will be better in cases of title and subtitle to use non abbreviated way of expression.

Ln233: Maybe it will be more appropriate to say: Numerous studies [15-17......49], included in current survey, have ...

Ln248: Echinacea needs to be in italics. However, if you say Echinaces, then it is a bit confusing. Maybe you need to express yourself better and say what kind of extract of Echinacea was applied or give some more specific description.

Ln250: Specify what Kan Jang is. Is this commercial products? Please, provide origin of the product.

Ln264: sidoides supposed to be with not capital s.

All case studies under 3.3.1. will beneficial if can be presented with a bit more details.

Providing examples for application of different plants and other extracts/products are interesting, however, authors will need to provide a bit more information. Maybe organizing a table with all these results will be good illustrative material as well.

Ln352: Maybe it will be appropriate to list what references were applied in this section when you have stated that "articles have been identified..."

Section 3.7.1. need a bit more details and examples. If I am not wrong, some homeopathic products are even on the market regarding flu.

In my opinion discussion can be extended a bit. Maybe a more critical view on the existing facts will be positive for the purpose of the manuscript.

Please, in conclusion, try to give a better defined take home message. Please, remove section 6. Patents. This is not relevant to the current manuscript.

Similar for section Supplementary materials

Manuscript have several figures and table (Presume supplementary) that where not cited into the text.

References needs to be formatted according to the regulations from Publisher and Journal.

Author Response

REVIEWER 3

The current review is interesting and gives some information about homeopathic and possible control in human (and why not veterinary) medicine. In my opinion paper can be suggested to the Editor to be considered for publication, however, authors will need to upgrade some parts, adjust other and maybe provide some additional information

Authors answer:   Thank you for your comments and the opportunity to improve this manuscript. We hope that the corrections made will make the manuscript more understandable and allow it to be published as soon as possible.

Some specific comments:

I am not sure if the format is really appropriate and it is according to the recommendations form the Journal and Publisher. On my knowledge Foods/Nutrients do not require splitting the abstract in subsections. Please, the technical team and the Editor of the Journal, can you please, give your opinion on this issue?

Authors answer:  Thank you for your comment. We downloaded the Nutrients template and reported the manuscript according to it. In this template, the abstract was subdivided into different subsections (background, methods, results, and conclusions).

It is not clear, paper coming as submitted to Nutrients, however, using template of Foods?

Authors answer:  Thank your for your comment. Initially, our scoping review was submitted to Foods, but the manuscript was transferred to Nutrients directly by the Editorial team with our approval, since Nutrients aims fit better with this study.

I am not sure if the format is really appropriate and it is according to the recommendations form the Journal and Publisher. Please, the technical team and the Editor of the Journal, can you please, give your opinion on this issue?

Authors answer:  Thank you for your comment. We downloaded the Nutrients template, and we respected the guidelines of this journal.

Ln36-37: Do not need quotation marks for the keywords

Authors answer:  Thank you for your comment. Done.

Ln58: In this context, by national, do you mind Italian? Please, specify. In fact, provided definition is Italian or EU agency? Please, specify

Authors answer:  Thank you for your comment. At L59, we modified “according to the European definition”. Done.

Ln63: Please, use opening citations marks for pathos. Needs to be "pathos".

Authors answer:  thank you for your comment that was addressed in the revised manuscript.

Section 91-97, it is not really needed to justify what original idea was and why authors have different types of paper builders

Authors answer:   thank you for your comment. We kept this statement for transparency since PRISMA guidelines require to report any deviation from the original protocol.

Approaches for the screening for appropriate literature and analysis of the recorded informative sources is well described and presenting authors strategy in preparation of current manuscript

Authors answer:   Thank you for your comment.

In my opinion authors describing with a bit over detail how selected studies were choice to be included in analysis, why is important to look not only at title and abstract, but that is also a kind of obvious criterium to read entire paper if you will use it as reference (however, some colleagues think that only checking abstract is sufficient, this is philosophical comment, nothing with the authors of the current paper). In my opinion, this part of the manuscript may be a bit reduced and later given a bit more focus on the examples of different homeopathic products and their role.

Authors answer:   We agree that the description of the study selection process may appear somewhat detailed, as the procedure of moving beyond title and abstract screening to full-text review is indeed a standard practice. Following your suggestion, we have streamlined this section to make it more concise and less self-evident in particular literature search and data extraction description. In parallel, we have expanded the discussion on examples of different homeopathic products.

Ln232: If I am not wrong, abbreviation UTTIs is for the first time used in this place. Please, in this and similar cases use full name before you introduce abbreviation, even for the well-known abbreviations. Moreover, maybe it will be better in cases of title and subtitle to use non abbreviated way of expression

Authors answer:  thank you for your comment that was addressed in the revisd manuscript.

Ln233: Maybe it will be more appropriate to say: Numerous studies [15-17......49], included in current survey, have ...:

Authors answer:  thank you for your comment that was addressed in the revised manuscript.

Ln248: Echinacea needs to be in italics. However, if you say Echinaces, then it is a bit confusing. Maybe you need to express yourself better and say what kind of extract of Echinacea was applied or give some more specific description:

Authors answer:  thank you for your comment that was addressed in the revised manuscript (L250).

Ln250: Specify what Kan Jang is. Is this commercial products? Please, provide origin of the product:

Authors answer:  thank you for your comment. We added “®” at L252.

Ln264: sidoides supposed to be with not capital s:

Authors answer:  thank you for your comment. Done (L266).

All case studies under 3.3.1. will beneficial if can be presented with a bit more details

Authors answer:   thank you for your comment. We conducted a scoping review, not a systematic review; for this reason, the aim was to provide a comprehensive, 360° overview of the state of the art of the literature concerning the use of phytotherapeutic and homeopathic remedies in the treatment of acute pediatric infections.

Providing examples for application of different plants and other extracts/products are interesting, however, authors will need to provide a bit more information. Maybe organizing a table with all these results will be good illustrative material as well.

Authors answer:  thank you for your comment. The new Table 2 report information on this issue.

Ln352: Maybe it will be appropriate to list what references were applied in this section when you have stated that "articles have been identified...":

Authors answer:  Thank you for your comment. L355: done.

Section 3.7.1. need a bit more details and examples. If I am not wrong, some homeopathic products are even on the market regarding flu.

Authors answer:  Thank you for your comment. We conducted a scoping review, not a systematic one; for this reason, the objective was to provide a general, comprehensive overview of the state of the art of the literature regarding the use of homeopathic and phytotherapeutic remedies in the treatment of acute pediatric infections. Moreover, based on the inclusion criteria selected to screen the studies, we did not identify any manuscripts linking homeopathic products to the treatment of Flu in children.

In my opinion discussion can be extended a bit. Maybe a more critical view on the existing facts will be positive for the purpose of the manuscript:

Authors answer:   thank you for your comment. L416-L571. We extended the discussion and make more critical.

Please, in conclusion, try to give a better defined take home message:

Authors answer:   thank you for your comment. L573-L589. We modified the conclusion in order to make them more critical and to define the final message.

Please, remove section 6. Patents. This is not relevant to the current manuscript.

Authors answer:   thank you for your comment. Section 6 “Patents” cannot be removed, as it contains information related to funding, author contributions, and conflicts of interest.

Similar for section Supplementary materials. Manuscript have several figures and table (Presume supplementary) that where not cited into the text:

Authors answer:   thank you for your comment. We modified the text and added all supplementary materials in the text. Also, we removed the section about “Appendix 1 – Supplementary Material”

References needs to be formatted according to the regulations from Publisher and Journal:

Authors answer:   thank you for your comment that was address in the revised manuscript.

Round 2

Reviewer 1 Report

Comments and Suggestions for Authors

nutrients-3855498
Manuscript: "Phytotherapic and Homeopathic Interventions for Pediatric Infections: A Scoping Review"
---
The article has been significantly improved. But there are still pending tasks:
#Major
-Clearly distinguish phytotherapy from homoeopathy in the introduction and results.
-Specify inclusion/exclusion rationale for bee products (sometimes described as nutraceutical, sometimes phytotherapeutic).
- Provide full search strings for all databases in the supplementary material.
-Clarify inter-rater reliability (e.g., Cohen’s kappa) during screening and data extraction.
-Expand discussion on heterogeneity of interventions (dosage, formulations, treatment duration).
-Explicitly discuss the clinical significance vs statistical significance of outcomes.
-Highlight whether results are robust enough for clinical recommendations.v
-Conclusions should state more explicitly that evidence for homeopathy is weak and inconsistent, and that current findings cannot support its recommendation.
- For phytotherapy, emphasize that promising results remain limited to URTIs and bronchitis, and for other infections there is a lack of evidence.
#Other issues
-Abstract: Refine conclusion. Currently it is too general. It should specify “evidence is limited to URTIs/bronchitis and does not extend to other pediatric infections.”
-Figures 2 and 3 (risk of bias) should be clearer (improve resolution and labeling).
- Consider summarizing key interventions and outcomes in a concise evidence table (beyond Table 2).
-Ensure that all references are formatted according to Foods guidelines.
- There are some minor inconsistencies (i.e., punctuation, author initials, spacing).
-Check out that the latest studies up to April 2025 are included (update information if new trials have been published since the last search).
-Minor grammar issues (e.g., “pf phytotherapeutic”; change to “of phytotherapeutic” in Table 3).
-Simplify some very long sentences in the Discussion section to improve clarity.
-Explicitly mention whether an ethical approval was required for this review (usually not, but journals often request a statement).

Comments on the Quality of English Language

Please, see report

Author Response

The article has been significantly improved. But there are still pending tasks:
Author answer: Thank you for your comments and the opportunity to improve this manuscript. We hope that the corrections made will make the manuscript more understandable and allow it to be published as soon as possible.

Major
-Clearly distinguish phytotherapy from homoeopathy in the introduction and results.
Author answer: Thank you for your comment. In the Results section, we have subdivided each paragraph into two distinct categories, namely phytotherapeutic and homeopathic treatments. In the Introduction, we have added further details regarding phytotherapy (L68–71) and homeopathy (L72–76). Moreover, we have introduced a “third” category including bee-derived products, since these cannot be classified as phytotherapeutics (given their non-plant origin) nor as homeopathic remedies.

-Specify inclusion/exclusion rationale for bee products (sometimes described as nutraceutical, sometimes phytotherapeutic).
Author answer: Thank you for your comment. Bee products, such as honey, propolis, and royal jelly, are considered a different and specific category, not as phytotherapeutic and homeopathic treatments, since they do not have a plant origin, nor as homeopathic, as they are not obtained through dilution and dynamization processes. For this reason, honey, propolis, royal jelly, and other bee products are commonly referred to within the framework of apitherapy, which represents a distinct category (Fratellone PM, Tsimis F, Fratellone G. Apitherapy products for medicinal use. J Altern Complement Med. 2016;22(12):1020-1022. doi: 10.1089/acm.2015.0346).

- Provide full search strings for all databases in the supplementary material.

Author answer: Thank you for your comment. Research strings are in the supplementary materials (Table A1).

-Clarify inter-rater reliability (e.g., Cohen’s kappa) during screening and data extraction.

Author answer: Thank you for your comment. In the Materials and Methods section, we added a new paragraph (2.4), titled “Inter-rater agreement”, describing the use of Cohen’s kappa statistic to assess the level of agreement between the two reviewers (L181–L186). In the Results section, we reported the calculated level of agreement, which indicated a substantial concordance between reviewers.

-Expand discussion on heterogeneity of interventions (dosage, formulations, treatment duration).

Author answer: Thank you for your comment. In paragraph 4.5 (L573–582), we have added references to the heterogeneity of the studies in terms of methodology, dosages, types of intervention, categories of active compounds, populations, and outcomes.

-Explicitly discuss the clinical significance vs statistical significance of outcomes.

Author answer: Thank you for your comment. As this is a scoping review, our aim is not to identify statistically significant differences. Rather, the objective is to collect and map the available scientific literature regarding the potential efficacy of homeopathic, phytotherapeutic, and bee-derived products in the treatment of acute pediatric infections.

-Highlight whether results are robust enough for clinical recommendations.

Author answer: Thank you for your comment. Building on the previous recommendation, our aim (as stated in the Methods and Discussion sections) is to identify potential positive trends regarding treatment efficacy, rather than to draw definitive or statistically significant conclusions about alternative treatments to conventional drugs (e.g., antibiotics).

-Conclusions should state more explicitly that evidence for homeopathy is weak and inconsistent, and that current findings cannot support its recommendation.

Author answer: Thank you for your comment. We have revised the conclusion (L584-597).

- For phytotherapy, emphasize that promising results remain limited to URTIs and bronchitis, and for other infections there is a lack of evidence.

Author answer: Thank you for your comment. We emphasized this aspect in paragraph 4.4, about the gaps in the current literature (L541-552).

#Other issues
-Abstract: Refine conclusion. Currently it is too general. It should specify “evidence is limited to URTIs/bronchitis and does not extend to other pediatric infections.”

Author answer: Thank you for your comment. We have revised the conclusions (L35-L41).

-Figures 2 and 3 (risk of bias) should be clearer (improve resolution and labeling).

Author answer: Thank you for your comment. We have attempted to improve the labeling and resolution aspects; however, this represents the best result currently achievable.

- Consider summarizing key interventions and outcomes in a concise evidence table (beyond Table 2).

Author answer: Thank you for your comment. We chose to present the outcomes of the individual studies within the Results section, while Table 1 includes only the study characteristics. We considered the outcomes as part of a more descriptive approach, and it seemed clearer to leave them in the Results section rather than attempting to summarize or schematize them.

-Ensure that all references are formatted according to Foods guidelines.

Author answer: Thank you for your comment. All references are formatted according to Nutrients guidelines.

- There are some minor inconsistencies (i.e., punctuation, author initials, spacing).

Author answer: Thank you for your comment. We have revised all references and corrected author initials, spacing, and punctuation.

-Check out that the latest studies up to April 2025 are included (update information if new trials have been published since the last search).

Author answer: Thank you for your comment. We conducted a thorough search and did not identify any new data or studies on this topic published since April 2025.

-Minor grammar issues (e.g., “pf phytotherapeutic”; change to “of phytotherapeutic” in Table 3).

Author answer: Thank you for your comment. We have revised Table 3.

-Simplify some very long sentences in the Discussion section to improve clarity.

Author answer: Thank you for your comment. We have revised discussion (L491-582).

-Explicitly mention whether an ethical approval was required for this review (usually not, but journals often request a statement).

Author answer: Thank you for your comment. Ethical approval was not required because our data were already published and do not involve original research.

Reviewer 3 Report

Comments and Suggestions for Authors

In my opinion authors have corrected the manuscript according to the critics and recommendations form reviewers and can be suggested for publication.

Author Response

Dear reviewer, 

Thank you for your comments. We hope that the manuscript could be more understandable.